# Household storage, surplus and supra-household storage in prehistoric and protohistoric societies of the Western Mediterranean

**Georgina Prats**[1,2]*, **Ferran Antolín**[1], **Natàlia Alonso**[2,3]

**1** Integrative Prehistory and Archaeological Science (IPNA/IPAS), Department of Environmental Sciences, University of Basel, Basel, Switzerland, **2** Departament d'Història, Facultat de Lletres, Universitat de Lleida, Lleida, Catalonia, Spain, **3** Departament d'Història, Facultat de Lletres, INDEST, Universitat de Lleida, Lleida, Catalonia, Spain

☯ These authors contributed equally to this work.
\* georgina.pratsferrando@unibas.ch

**Data Availability Statement:** All relevant data are within the manuscript and its Supporting Information files.

## Abstract

The objective of this paper is to assess foodstuff storage throughout Recent Prehistory (5600–50 BCE) from the standpoint of the three different types (household, surplus and supra-household) identified in the northeast of the Iberian Peninsula. The volumetric data of the underground silos serves as a proxy to evaluate the link between them and the agricultural systems and technological changes. The study also assesses the ability, and specifically, the will of the ancient communities of the northeastern Iberia to generate domestic and extra-domestic surpluses.

## Introduction

François Sigaut a few decades back wondered whether the academic interest in storage techniques among ancient communities stemmed from a deliberate awareness of their value or simply represented a passing fashion [1] (p. 61). Since then, research in storage systems has developed in both archaeology and ethnography. The intent of this research from the academic standpoint has been to gain knowledge in field of palaeoeconomics while from the framework of the United Nations their study has served to implement traditional storage techniques in areas bereft of modern technology [2–18]. Sigaut's modest reflection in the early 1980s is now a thing of the past as the archaeological record is placing more and more importance on basic concepts such as that the duration of foods is limited and that surpluses require storage for consumption until the next harvest, to provide the seeds for the next sowing, as well as to be exchanged. Thus, the main objective of storage systems was long term preservation with minimal loss [19]. These features were highly relevant solutions and an important strategy within the agricultural and productive system of the past communities. Hence, variables linked to the storage of agricultural products (techniques, structure capacity, management, distribution, typology, etc.) must be integrated into socio-economic analyses of early agricultural societies

**Funding:** This research has been funded by the Swiss National Science Foundation (http://www.snf.ch/) as part of a SNF Professorship (PI: F. Antolín), grant number: PP00P1_170515, and by the Spanish Ministry of Science, Innovation and University projects HAR2008-0526 and HAR2016-78277-R. The funders had no role in study design, data collection and analysis, decision to publish, or preparation of the manuscript.

**Competing interests:** The authors have declared that no competing interests exist.

[8, 9, 11, 20–25] as they are the most reliable indicator of agricultural productivity, particularly in contexts where written records are unavailable.

Numerous strategies throughout the history of humanity were developed to ensure a deferred consumption of foodstuffs serving to guarantee the survival of the group in both hunting-harvesting [26, 27] as well as farming contexts [28] (p. 3). Resorting to features of conservation in archaeology is considered as an essential and inherent pre-adaptative stage prior to the adoption of agriculture [29–31 (p. 2)] [32, 33]. Storage practices, like grinding [34], therefore predate agriculture. Moreover, agriculture became possible as storage techniques allowed consumption throughout the year, thus bolstering the economic viability of sedentarisation [35]. Moreover, the development and transformation of storage methods over time can be linked to profound changes in ancient economy, settlement patterns, social relationships and demography [23, 36, 37]. Storage on many occasions is identified as key to the analysis of social inequality and is behind a surge of social complexity, the emergence of an elite and socio-economic inequality [38 (p. 3), 39]. Both domestic and surplus management, in fact, is fundamental to any stratified hierarchical society [40].

A domestic unit from the anthropological point of view is understood not only as a reproductive group but as an element of economic and social cooperation [41] which produces and provides its members with the resources necessary for survival. This type of unit cooperates and shares a number of economic activities on a daily basis: production, consumption, resource pooling, distribution, transmission, co-ownership, reproduction and shared ownership [42 (pp. 620–621), 43 (p. 6)]. The nuclear unit, in turn, represents a group of 5–7 economically autonomous individuals and extensive unit is represented by a greater number [44, 45 (p. 121)].

Among domestic units this analysis also distinguishes *individual/private* storage for own use and consumption and *collective/communal* storage by an extended group either under individual or collective control with the intention of a use of the stored product not depending exclusively on the will of the domestic unit. This is an essential element at times serving as the focus of archaeological research often based on criteria such as the physical location of the storage features either in a central or domestic space of the settlement, or in spaces of difficult access such as caves [46] (p. 83). A second criterion is that of structure capacity as the volume of certain storage features exceeds by far that of the production of a domestic unit. This reflects an alternate use and/or a level of social hierarchy as the accumulation of wealth of certain individuals is most often linked to an agricultural surplus [47–50]. This specialisation or hierarchy is also observed among settlement types [51, 52] as these can assume specific functions according to whether they served as production centres and accumulators or distributors and consumers [53 (p. 136), 54 (pp. 13–26)]. Other pre-requisites such as road networks, appropriate transport systems and institutional structures providing security to the networks [50] must be in place in order to attain this level of specialisation.

In order to delve deeper into the understanding of the different scales or levels, this study defines the following three main storage categories: household, surplus and supra-household.

## Household storage

This category is linked to subsistence practices of a household including a private storage of food for daily use. This therefore combines short-term and long-term storage techniques with features of different capacity and location within the dwelling, farm or village, etc. The features guarantee the seeds necessary for the next sowing, which can take place, depending on the season, from 3 and 8 months after the harvest. To assure their germination, the seeds require excellent preservation conditions [55].

There are various theoretical approaches as to the average production capacity of a domestic unit (see S1 Text), that is, the amount of grain required to sustain a family for a year. Ethnographical research by Kramer [45] establishes the volume at 1,000 litres (1 m3), that is, almost a ton of grain [45 (p. 37), 56 (p. 49)]. Halstead's [37] (p. 162) study of Greek farmers suggests a volume corresponding to 1–1.5 tonnes, including retaining a margin for possible losses. This equates with a storage capacity of 1,300 and 3,000 litres depending on the moment of the product's processing. Similar values are advanced by Sigaut [57] (p. 165) for preindustrial societies. The volume of domestic grain storage, in fact, largely depends on the agricultural model and the level of technological innovation [42].

## Surplus

This relates to an excedentary production, that is, a household production beyond its annual immediate needs [26]. It can stem from favourable weather conditions, higher labour capacity, or simply a more efficient agricultural technology. Surpluses serve as risk-reducing strategies in anticipation of poor harvests [58] (p. 598), as a sort of "social storage" intended for exchange [36], for immediate sharing in form of feasts [58] (p. 600) or for purposes of trade/speculation.

Surpluses are generally considered to stem from agricultural expertise. The findings of Halstead [36] demonstrate, nonetheless, that surpluses are intrinsic the productive model. An increase of surplus capacity is not infinite in nuclear families or in non-mechanised contexts as the workforce is one of the major limiting factors [37, 39]. Thus, a means to increase the volumes of a surplus is to multiply the number of household unit members [42 (p. 622)].

## Supra-household storage

This category relates to the collective/communal storage cited above, in particular to features that contain more than the yield of a household intended for either collective use or for accumulation/speculation. The emergence and development of this storage category is intrinsically linked to social superstructure, that is, with political and institutional organisation and with the existence of stable commercial networks [15, 52, 53].

The focus of this study is that of the underground silo (Fig 1). There is a wide consensus that these features served as containers to preserve plant products, especially cereals [8 (p. 206), 59–62 (p. 76)]. This function is confirmed by multitude ethnographic studies [46, 55, 63], classical and historical texts ([64], Book I, LVII, 2; [65], Book XVIII, 306–307) and by experimental archaeology [60, 66]. Moreover, these features are very common, albeit not exclusive, throughout the Mediterranean Basin [28]. Due to their favourable conditions of conservation, they are particular well-known at open-air sites [67].

Furthermore, among the different storage systems, the underground silo is the only type that can be approached on a broad scale from quantitative perspective. It must be noted that the study area, the NE of the Iberian Peninsula, is devoid of Tell type settlements known for other types of storage systems notably the "bin type" features [39, 68, 69] that have been the subject of research adopting a similar approach to that of this study.

## Underground storage capacity as a proxy of agricultural productivity

The storage capacity is an indispensable variable in the study of the production and surplus of past societies [63]. Thus, the premise of the current study is that domestic and extra-domestic storage and productive surpluses are tangible in the archaeological record and can be quantified by measuring silo volumetric capacity. Moreover, capacity values directly reflect potential functions and motives behind storage (consumption, sowing, poor harvests, exchange, trade,

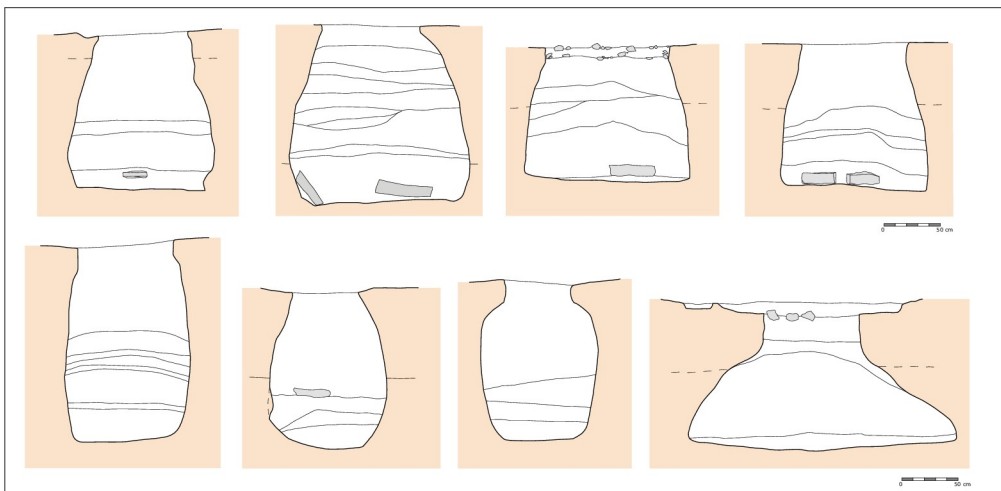

**Fig 1. Some examples of Early Bronze Age silo pits of Minferri site (2300–1300 cal BCE) (from Fig 18 in Prats 2013 and modified by Georgina Prats).** Republished form [REF] under a CC BY license, with permission from [Revista d'Arqueologia de Ponent], original copyright [2013].

etc.). Silo capacity has also often served a) to define which structures among family units functioned as reserves of products; b) to infer an extra-domestic character of a community's reserves; c) to approach the question of the main economic strategy of a community; d) to identify the role and level of significance of these features in agriculture; and e) to define the index of productivity, the extension of arable land, demographics, etc. Thus, analyses of silo capacity can shed light on socio-economic aspects of ancient communities.

One of Gerhard Bersu's main contributions to the study of storage capacity was to estimate the number of acres required by an agriculturally community to render them viable and self-sufficient [70]. Thus, research in this field has tended to establish a direct relationship between storage capacity and immediate field production. This association, reinforced by ethnographic studies, emphasises that the storage structure volume is directly linked to the yield of each harvest and the dimension of cultivated surfaces. According to Alonso *et al.* [16] (p. 47), the volume of the subterranean silos of the Ouarten Berber tribe in Tunisia depended on the number of hectares available to the *producer/s* and the amount of production. The same was observed for the communities in the Rif Mountains of Morocco where Peña-Chocarro *et al.* [14] (p. 383) recorded that the dimensions of storage features varied depending on the amount of cereal to store. The structures usually ranged between 3 m in width and 5 or 6 m in depth tantamount to a capacity of up to 1,000 kg. Moreover, there is an intrinsic association between storage features and the notion of surplus. Hence when there are silos, there is a surplus, and, if these features are numerous, they fall into the category of *silo fields* (see S1 Text).

But how can silo capacity to be interpreted in a changing socio-economic context? What is the average capacity expected of nuclear and extended families? How is silo volumetric variability explained? The average volume of the Neolithic houses of Çatalhöyük (6250–5850 BCE, Turkey), for example, is about 1,000 litres [68] (p. 661), values analogous to those in the Iberian Peninsula on more recent Bronze and Iron Age contexts [8 (p. 231), 25, 71 (p. 141)]. Vaquer [46] (p. 83) also notes that Protohistoric family silos hold less than 3,000 litres while those of larger volumes are collective, values that can be applied, in turn, to the Iberian (Iron Age) *oppida* of northeastern Iberia [72] (p. 39). Hence, the volume of production of domestic units does not coincide from one socioeconomic context to another. This implies the need of resorting to revisable theoretical parameters in order to interpret storage structure capacity, a notion

bolstered by a recent study on the evolution of the average and maximum capacity of underground silos ranging from the Neolithic to Romanisation of NE of the Iberian Peninsula [25, 63].

The aim of this study is therefore to define these three types of storage features throughout the Recent Prehistory in the NE of the Iberia by analysing the volumetric data of underground silos. The intention is to identify if volumetric capacity, from a global perspective, allows to identify intra-site growth or reduction. Moreover, the intention of this study is to attempt to assess the capacity, and more specifically, the will of these ancient communities to generate both domestic and extra-domestic surpluses over time. Finally, the study aims to evaluate the link between means of storage and technological changes, and the prevailing type of agriculture of these socio-economic systems.

This approach is nonetheless not devoid of interpretive and methodological limitations. For example, the archaeological contexts of these features (outside or inside dwellings, grouped or spread out within dwellings, beyond or within settlements, near and/or around settlements, roads, ports, etc.), although essential, is complex to analyse in detail, and can lead to straying from the global perspective. This paper therefore avoids the site-scale analysis, which would vastly surpass its goals, and would encounter other problematics such as the scarcity of well-preserved settlements where i.e. dwellings and storage pits could be associated. This is a widespread phenomenon in the Western Mediterranean. Consequently single values will not be considered in this evaluation, keeping the focus on the general trend given by all features considered for each phase, as in Prats *et al*. [25]. That said, several aspects require highlighting.

First of all, the register of storage features is biased in terms of excavations. That is, sites linked to greater numbers of silos are often excavated during preventative urban interventions and therefore in spaces not selected deliberately based on scientific criteria. There is also the problem of structure conservation as they often suffer from a high degree of erosion truncating their top diameter leading to partial and often incomplete registration. While these factors are significant in the framework of small-scale studies (i.e. at the site level), they nonetheless do not hinder detecting general tendencies through more general survey with a broader geographic and chronological scope. Secondly, it is arduous to determine the useful life and number of silos functioning simultaneously at a site. Although this factor undoubtedly complicates land use modelisations, this does not affect the current study as its intention is not to calculate a site's cultivated area at any specific moment. Thirdly, not all of these features necessarily served to store grain, or were filled to the brim [8 (p. 207), 40 (p. 332), 73]. Hence a silo with a 1,000 litres capacity cannot necessarily be equated with 1,000 litres of grain. Fourthly, the weight of products in a silo can waver according to the type of cereal and if storage took place in the form of spikes, spikelets or whole grains [8] (pp. 202–203). In general, the study area of this paper is dominated by free-threshing cereals such as naked wheat (*Triticum aestivum/durum/turgidum*), naked barley (*Hordeum vulgare* var. *nudum*) and hulled barley (*Hordeum vulgare*)-, and storage in the form of spikes is unusual, only recorded at the Neolithic site of La Draga, where storage took place inside pile-dwellings [74, 75]. Finally, it is necessary to bear in mind that the current study's focus is regional and split up into broad chronological phases.

## Methodology

All the sunken structures dated between the Neolithic and the Roman period (5600–50 BCE) from the NE of the Iberian Peninsula (including those not identified by archaeologists as silos) were gathered into a database (Fig 2).

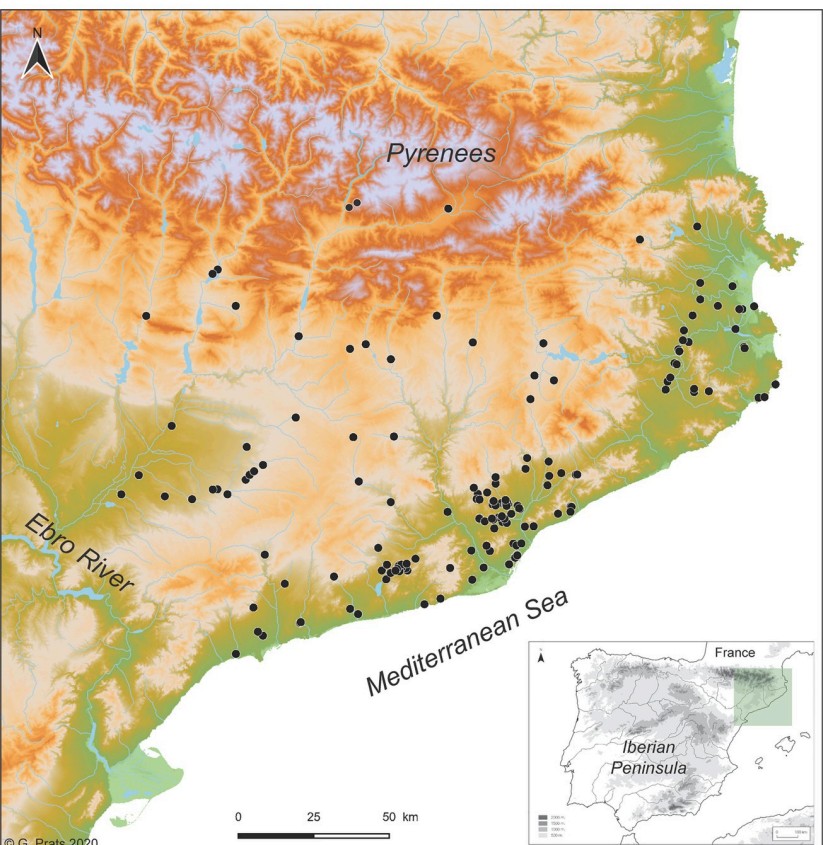

**Fig 2. Overview of the study area.** The black dots represent the archaeological sites cited in the study (Software: QGIS3.6, © European Union, Copernicus Land Monitoring Service [2016], European Environment Agency (EEA)).

## Resources and dataset

The data of this study were collected from published research and archaeological reports available at the Archaeological Service of the Generalitat of Catalonia. Other information was gleaned directly from excavation projects. These elements were registered in a database created with the application Filemaker, Inc [76]. The data fell into four complimentary categories: a) site (*site*), b) settlement phase (*site-phase*), c) function (*functional set*), and d) structure (*pit-silo*) [63]. The archaeological site category are broken down into a) open-air or fortified settlements; b) *silo field*; c) isolated silos; d) open air settlement + *silo field*; e) and fortified settlement + *silo field*. The settlements from these categories are represented in the S1 Table. Moreover, this study applied the traditional chronological phasing (Table 1) and all features devoid of clear chronocultural contexts (described as i.e. "prehistoric", "neolithic", etc.) were discarded.

This study applied an index based on the correlation between the top diameter of the structure (Ø) and its depth (D) (Ø/D) [77–81] in order to differentiate, by chronological phase, the deeper features from other pits. Applying the Ø/D index allows individualising certain tendencies and objectively identifying different pit categories characterised by inflections and slight variations of their sections. The four categories are illustrated in the charts of Fig 3. Thus, these features can be grouped into four broad categories according to their (Ø/D) index: deep pits (Ø/D index <2.5), simple pits (Ø/D index between 2.5 and 4.5), basins (Ø/D index between

**Table 1.  Total number of sites and silos for each chronological period, and the number of silos serving for the volumetric study.**

| CHRONOLOGICAL PERIODS | N. SITES | N. SILOS | N. SILOS—VOLUME |
|---|---|---|---|
| EN—Early Neolithic (5600–4500 cal BCE) | 15 | 50 | 42 |
| MN—Middle Neolithic (4500–3200 cal BCE) | 33 | 163 | 142 |
| LN—Late Neolithic—Chalcolithic (3200–2300 cal BCE) | 19 | 109 | 102 |
| EBA—Early Bronze Age (2300–1300 cal BCE) | 43 | 454 | 391 |
| LBA—Late Bronze Age (1300–750/700 cal BCE) | 27 | 299 | 119 |
| EIA—Early Iron Age (750/700–575/550 BCE) | 29 | 374 | 143 |
| EI—Early Iberian (575/550–450/400 BCE) | 9 | 76 | 41 |
| MI—Middle Iberian (450/400–200 BCE) | 52 | 370 | 293 |
| LI—Late Iberian (200–50 BCE) | 73 | 537 | 391 |
| TOTAL | 300 | 2432 | 1664 |

4.5 and 6.5) and lenticular pits (Ø/D index ≥ 6.5). Other studies have likewise defined these categories (e.g. [79]). However, similar categories do not necessarily always bear the same intervals as they depend on the type of pit. This criterion is descriptive rather than functional. Thus, certain structures identified as deep pits did not necessarily serve as silos, and that features falling into the basin or lenticular categories were assigned as deep pits. In any case, the index has served to globally differentiate the deep pits or silos serving as the base of this study (S2 Table).

Based on these criteria, silos bear a Ø/D index that lines up with the deep pit category. However, it was necessary to pass certain features through a second filter. These consist of structures with a depth exceeding 35 cm [82] (p. 148) and an index between their depth (D) and top diameter (Ø) greater than 0.7 [8, 81, 83 (p. 19)]. This D/Ø index is nonetheless only applied to features starting with the Early Bronze Age because if applied to the Neolithic structures most would have to be discarded.

## Calculating silo-pit capacity

Calculating structure volumetric capacity is based on their conserved archaeological morphology. The calculations are therefore an underestimation of their real storage capacity [68] (p. 661).

Different methods have been applied throughout historiography to compute the volume of structures [8 (p. 216), 48, 63, 84–88]. These methods are based either on geometric formulas and a solid of revolution representation (see Prats [89] for the comparison of the systems). The current study opts for the first of the methods as the intention is to carry out a macro-spatial analysis. This method, common in archaeological literature, starts with a morphological analysis of each structure so as to define a geometric form (Table 2) and from there, based on the appropriate formula, calculate the volume.

## Graphic representation of silos

The graphic representation of the data is based essentially to three elements: box plots, violin plots, histograms and geographic maps. Box plots (RStudio v.1.2.1335) [90] serve to represent the volumetric diversity of each chronological phase. The basic premise is that the dispersion of the different types of storage structures (household, surplus and supra-household) can be reflected in a quantitative manner. The box plots method also serves to represent the standard deviation of pit volume at the level of the sites (sites with 3 or more of these structures) so as to determine if there is an increase or decrease of diversity within the same settlement throughout

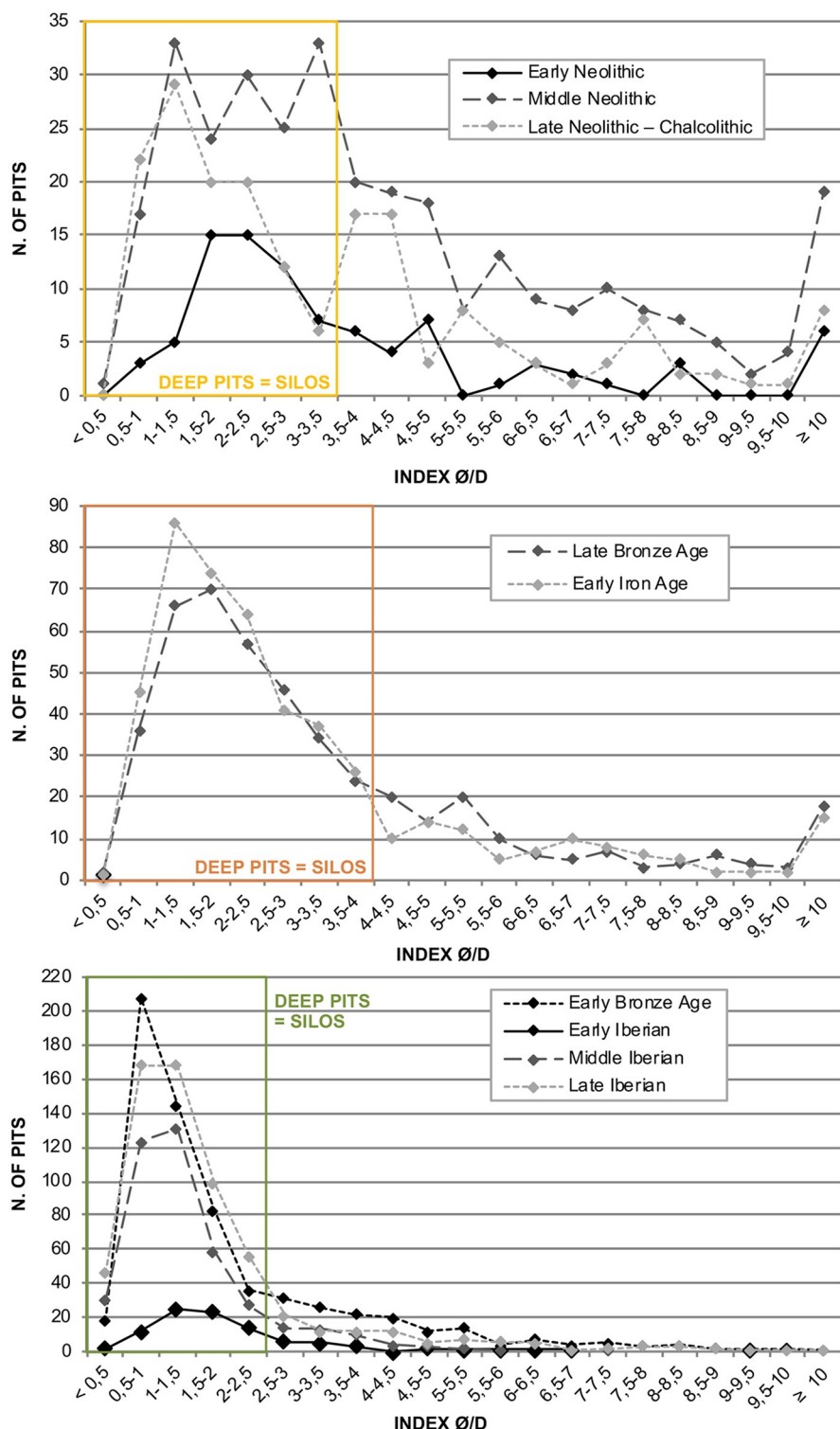

**Fig 3. Line charts by chronological phase of the diameter/depth indexes that distinguish deep pits (= silos) from other pits (see S2 Table).**

**Table 2. Formulas serving to calculate silo volume according to their morphology.**

| TYPE OF SILO | FORMULA SERVING TO CALCULATE SILO VOLUME |
|---|---|
| BELL-SHAPED | [3,14* depth)/3*((Sup. radius*Sup. radius)+(Inf. radius* Inf. radius)+Sup. radius*Inf. radius)/1000] |
| SPHERICAL & GLOBULAR | [(maximum ø -(Sup. radius.*Sup. radius)/4)+(Sup. radius*Sup. radius)*3,14* depth)/1000] |
| FUNNEL | [0,785398* depth*(Arithmetic mean ø + 3/5*(maximum ø—arithmetic mean ø))*(arithmetic mean ø + 3/5* (maximum ø—arithmetic mean ø)))/1000] |
| BOTTLE | [3,14*(Sup. radius*Sup. radius)* depth)/1000] |
| CYLINDRICAL | [3,14*(Sup. radius*Sup. radius)* depth)/1000] |
| BICYLINDRICAL | [0,785398* depth*(arithmetic mean ø + 3/5*(maximum ø—arithmetic mean ø))*(arithmetic mean ø + 3/5* (maximum ø—arithmetic mean ø)))/1000] |
| ELLIPSOIDAL | [(maximum ø-(sup. radius*sup. radius)/4)+(Sup. radius*Sup. radius)*3,14* depth)/1000] |
| BICONCAVE & PYRIFORM | [0,785398* depth*(arithmetic mean ø + 3/5*(maximum ø—arithmetic mean ø))*(arithmetic mean ø + 3/5* (maximum ø—arithmetic mean ø)))/1000] |
| DIVERGENT BELL-SHAPED | [3,14* depth)/3*((Inf. radius*Inf. radius)+(Sup. radius*Sup. radius)+Inf. radius*Sup. radius)/1000] |
| BICONICAL & TWIN-BELL SHAPED | [0,785398* depth*(arithmetic mean ø + 3/5*(maximum ø—arithmetic mean ø))*(arithmetic mean ø + 3/5*(maximum ø—arithmetic mean ø)))/1000] |
| HEMISPHERIC & SUBSPHERICAL | [3,14*(Sup. radius*Sup. radius)* depth)/1000] |
| TRAPEZOIDAL | [3,14*(Sup. radius*Sup. radius*Sup. radius)*2/3)/1000] |
| LENTICULAR | There is no formula |

the different chronological phases. Violin plots (RStudio v.1.2.1335) resemble box plots with the exception that they go further and depict probability density of the data at different values [91]. To create both box plots and the violin plots we have used the following functions; ggplot2.violinplot and ggplot2.boxplot, belonging to the libraries ggplot2-library(ggplot2)- and easyGgplot2 -library(easyGgplot2)-. The histograms, in turn, serve to classify the storage structures into the four categories, always according to the range of volume of each chronological phase: small, medium, large and very large silos. This serves as the base of the argument of the different levels of storage, the objective of this study. Finally, the visual representation of the findings through geographic maps [92, 93] allows identifying the different patterns of spatial distribution of small, medium and large silos.

## Results

The filtering of the deep pits (DP) among the total assemblage of all the features has led to identifying a total of 1,651 silos from 172 different archaeological sites. Their chronology and settlement, as well as the calculation of their volume, are listed in the S3 Table.

The graph (Fig 4) reveals a progressive increase of volumetric capacity throughout the Prehistory and Protohistory in the NE of the Iberian Peninsula. Three chronological blocks clearly stand out: a) the Neolithic and the Early Bronze Age; b) the Late Bronze Age and Early Iron Age; c) and the Iberian period. The first reveals features with a storage capacity up to 1,300 L, a volume which can be practically englobed in the first quartile of the following blocks. On the other hand, the data within each of the phases of the Neolithic and Bronze Age point to a more homogeneous (more symmetrical) distribution than their later counterparts while the values of the mean and median from the Iron Age are progressively more distant. The Middle and Late Iberian period reveal greater differences between the extremes (minimum and maximum) and, therefore, are the periods with the most dispersed and variable data.

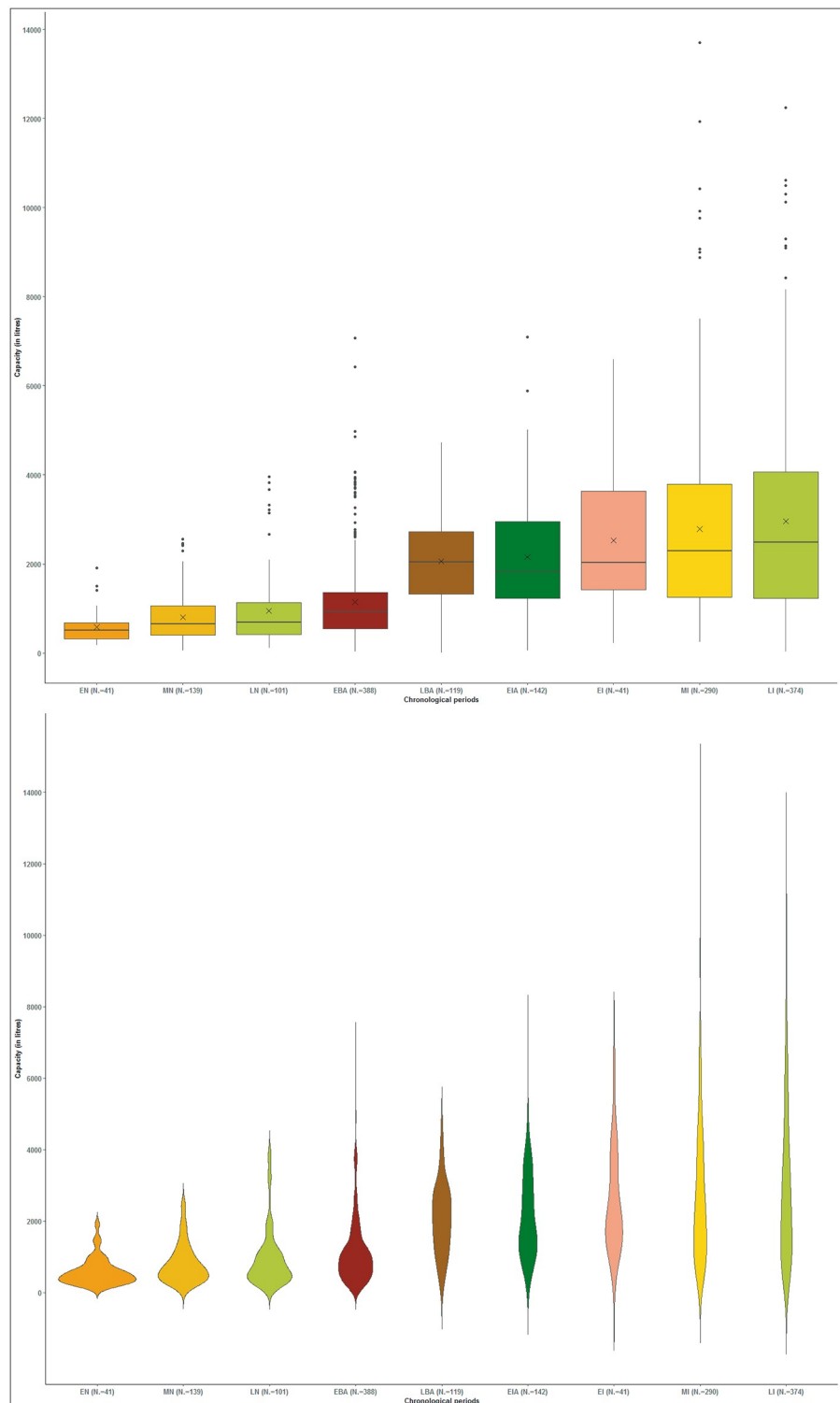

**Fig 4. Box plots and violin plots indicating silo volume capacity (in litres) by period.** The values of 45,953 L and 61,034 L of the Middle Iberian period and those of the Late Iberian period above 14,000 (15,945 L, 17,540 L and 22,810 L) were discarded.

The volumetric capacity of 50% of the structures in the Early Neolithic ranges between 319 and 677 L, in the Middle Neolithic between 385 and 1,041 L, in the Late Neolithic-Chalcolithic between 418 and 1,132 L and in the Early Bronze between 540 and 1,362 L. The capacity of this proportion of silos in the Late Bronze Age is between 1,320 and 2,719 L, in the Early Iron Age between 1,234 and 2,942 L, in the Early Iberian between 1,414 and 3,629 L, in the Middle Iberian between 1,246 and 3,787 L and in the Late Iberian between 1,240 and 4,061 L. There is therefore a clear increase of storage capacity throughout these periods.

Outliers appear throughout the entire Neolithic and the Early Bronze Age that are beyond the maximum range. But it is not until the Late Neolithic-Chalcolithic, and particularly the Early Bronze Age, that these values become much greater and really stand out. Thus, both maximum values and outliers increase during the Late Neolithic and the Early Bronze Age corresponding to volumes respectively of around 4,000 and 7,000 litres.

The Late Bronze Age, unlike the Early Bronze Age, does not appear to present outliers, that is, the increase of capacity in this time frame appears be homogeneous (25% between 2,719 and 4,729 L) (Fig 4). Outliers are likewise not recorded in the Early Iberian. Thus, the increase of the maximum capacities becomes homogeneous. A large number of outliers are observed, on the other hand, in the diagrams of the Middle and Late Iberian period. They stand out above the peaks of the two periods and are centred between 8,000 and 14,000 L in the Middle Iberian, and at about 12,000 L in the Late Iberian period. In this sense the outliers would have increased even higher if the earlier volumetric values (45,953 L and 61,034 L of the Middle Iberian and 15,945 L, 17,540 L and 22,810 L of the Late Iberian) had not been discarded (Fig 4).

To delve deeper into the volumetric capacity analysis and identify the trend throughout all the chronological periods, the structures were depicted in a graph at intervals of 500 L (Fig 5). An important aspect is that those with a 500–2,000 L capacity are found throughout all the periods. However, their representation in the Neolithic and the Early Bronze increases to 1,500 L since they comprise about 50% of the total. Silo capacity of 2,500 L is clearly represented in the Late Bronze Age onwards with these values surpassing 80% of the cases. Finally, the capacity beginning at 4,000 L is characteristic of the Middle and Late Iberian periods represented by more than 70% of the cases.

However, the graphs do not identify intra-site volumetric variability, that is, they fail to discern if the values diverge because the sites differ from each other or due to a higher degree of variability between them. The calculation of the standard deviation of all the volumetric values of settlements containing at least three pits is illustrated in box plots by chronological phase in order to offer a broader overview (Fig 6). The clearest result is the minimum straying from the average standard deviation throughout the Neolithic and the Bronze Age until attaining the Early Iron Age where it increases considerably despite representing the graph's smallest phase. As of this point the minimum is maintained during the Iberian era. These last phases reveal more extended boxes representing a greater diversity in capacity inside the settlements. Settlements with a low standard deviations, when compared to the previous phases, remain significantly lower.

## Discussion

### Levels of storage from the Neolithic to the Iron Age and their link to settlement dynamics

The findings of this study clearly indicate that since the values of storage capacity change over time, these features cannot be interpreted outside of their socio-economic framework. It is for this reason that silos were grouped into small, medium, large and very large categories, always within the range of values of each phase (Fig 7) before evaluating them in their socio-historical

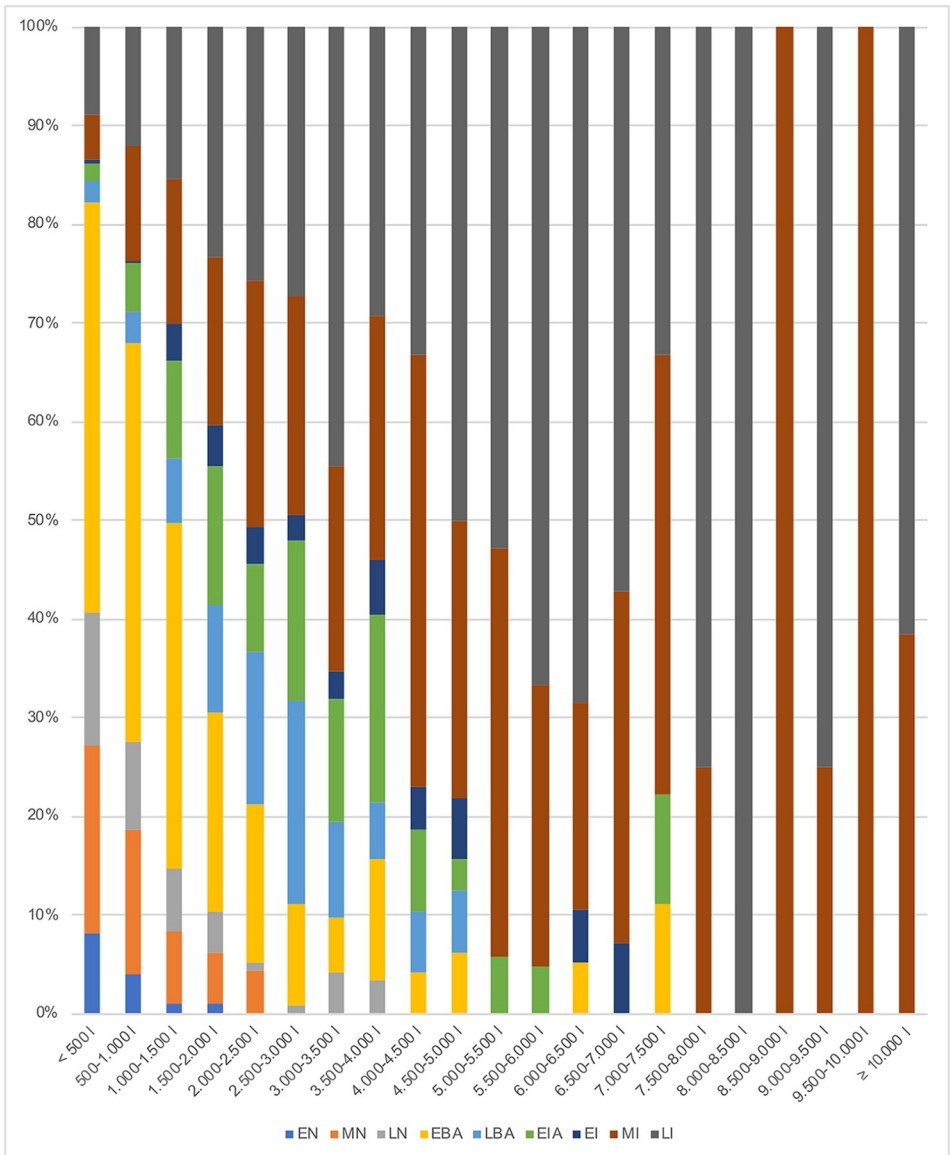

**Fig 5. Evolution of the minimum volumes calculated at intervals of 500 litres from the Early Neolithic to the Late Iberian period.**

and geographical context (Fig 8). All silos in the Early Neolithic, for example, would in fact fall within the category of small silos if they were in Iron Age or in the Iberian period contexts (<2000 L). Small silos are defined as storage features of a domestic unit with a modest capacity equivalent to the lowest level of variable inter-annual productivity. These values increase by approximately 300% in the chronological framework of this study. The values of medium silos are thought to represent the average domestic productivity with its natural surplus. The Iron Age sees a growth of 400% when compared to the Early Neolithic. The maximum values are the result of various factors such as exceptional crops (moment of economic growth, greater cultivation surface, etc.), collective pooling of seeds, or an accumulation intended for exchange. Finally, exceptional values, only known since the Bronze Age, reveal the ability of domestic units to accumulate reserves that are far beyond the productive capacity of a

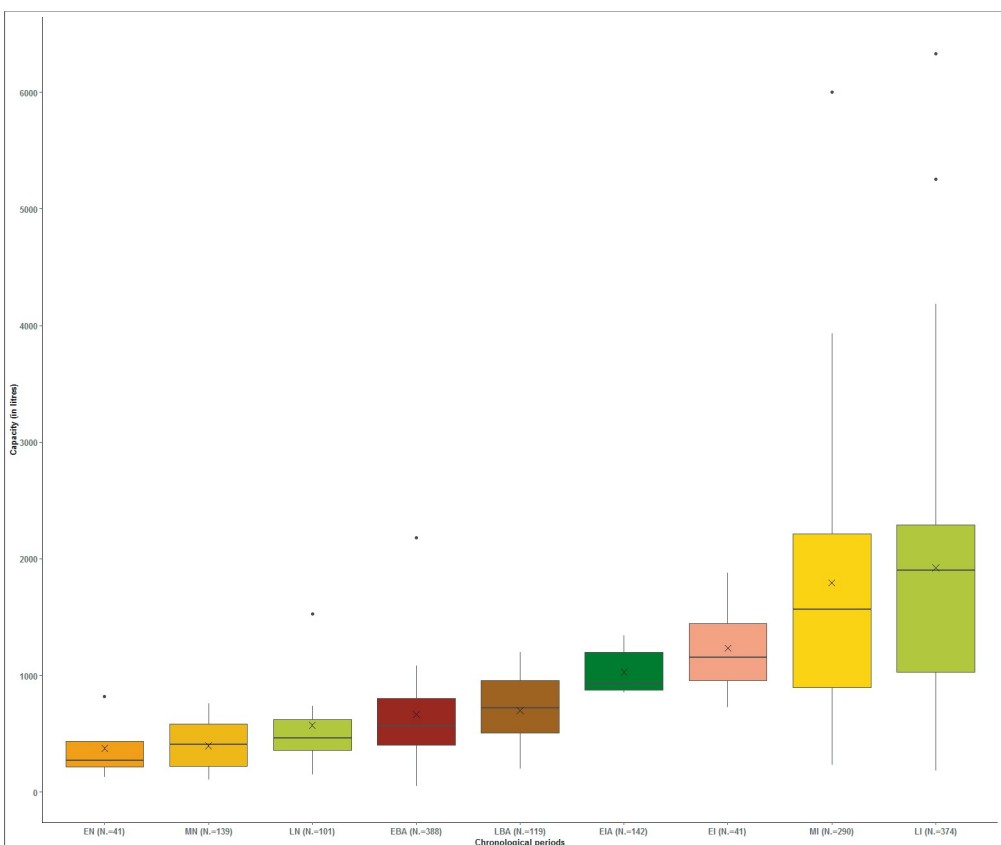

**Fig 6. Box plots indicating the standard deviation of all silo capacity by period (only settlements with more than three pits are taken into account).**

domestic unit. This development is particularly striking throughout the Iberian period and cannot be compared to the increases observed in earlier periods.

Throughout the Neolithic the number of small, medium and large silos doubles (100% increase) over the course of 3000 years (between 5500 and 2500 BCE). Although the Middle Neolithic is the period marked by the most sites with silos, it is the Late Neolithic that reveals a surge in the number and volume of silos per site [25]. Intra-site volume diversity remains constant throughout the three phases (Fig 6) probably due to a certain continuity of the social structure, productive model and political structure. There are nonetheless signs in the Late Neolithic of a transition toward a pattern that is characteristic of the Early Bronze Age. The NE of the Iberian Peninsula throughout all these phases appears to follow its own dynamic characterised by small and medium storage features and an absence of large and very large types (Fig 4). There are, furthermore, during the Early Bronze Age a modest number of settlements with a large number of domestic units that reach considerable dimensions [94–98]. These settlements are associated with many silos ranging in capacity from small to very large, a diversity that has yet to be recorded which probably stems from changes in the productive structure (more extensive) and social organisation (more hierarchical).

A transformation took place in the Late Bronze Age with the disappearance of underground silos in western Catalonia in the framework of the consolidation of proto-urban settlements away from the plains on high points [99, 100] leading to changes of economic strategy as use of storage systems [101]. As this new system did not resort to underground silos, it does not

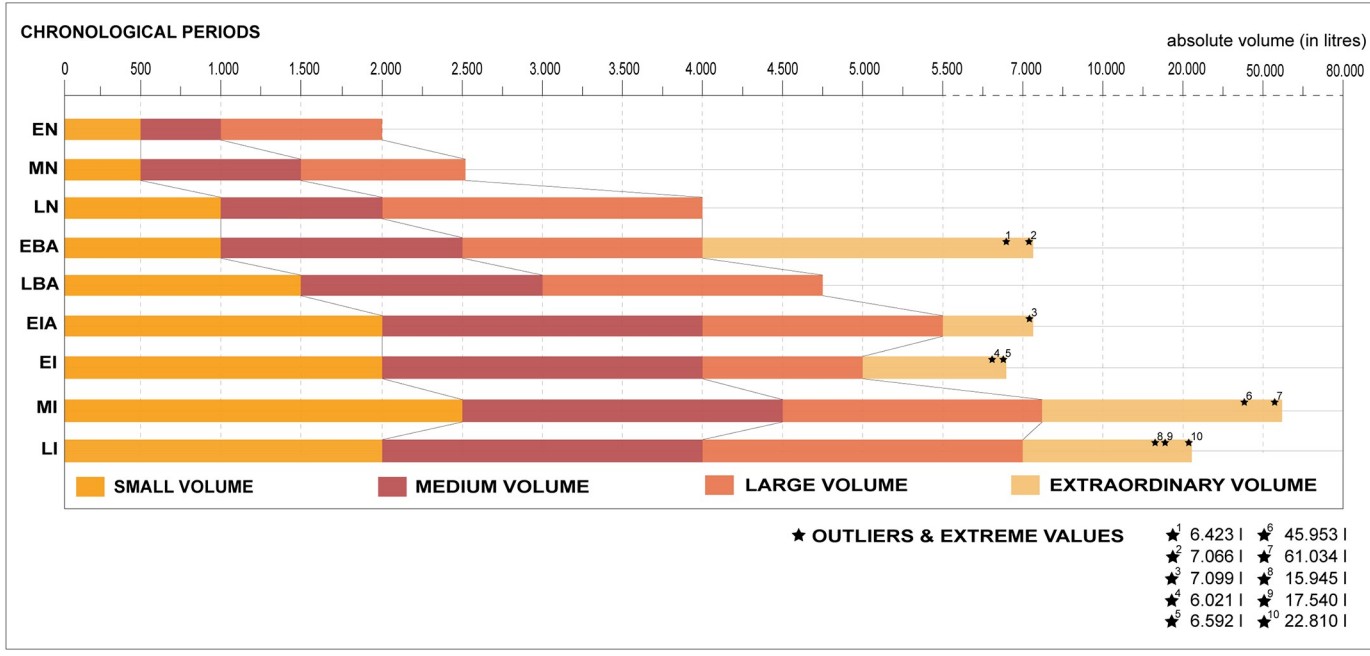

**Fig 7. Graphic representation of the evolution over time of silos of small, medium, large and very large volumes.**

form part of the current study. Eastern Catalonia, on the other hand, maintained these underground features—even more so than in previous phases—in loosely organised open-air settlements, a dynamic that persisted throughout the Early Iron Age. Indeed, silos are present at sites along the central strip of the Catalan coast (Fig 8) and appear as isolated pits or forming part of large concentrations (González *et al*. 1999, in [53] (p. 127)]. The standard deviation analysis indicates that a volumetric intra-site diversity increases in the Early Iron Age and there are no longer sites with very low values (in absolute terms) as in the Neolithic. It is at this time that silos linked to settlements comprising stone architecture began to appear in the framework of an incipient urbanism spawned by early contacts with Mediterranean populations from the western Phoenician colonies [53] (p. 128). In the case of western Catalonia, silos once again appear, although now of small capacity, (Fig 8), at proto-urban settlements [102, 103].

Domestic and surplus storage in the economic structure of the Iberian period mainly resorted to underground silos [53]. Although the archaeological evidence—perhaps limited— points to a reduction of the number of settlements with silos at the outset of the 6[th] century BCE (Early Iberian), this tendency changes radically during the Middle and Late Iberian periods. What now stands out is a solid occupation of the territory where silos are especially common to sites along the pre-littoral and coastal areas, as well as in the NE of Catalonia where their number is especially considerable [104–108] (Fig 8). The use of silos in the eastern part of the Western Plain and in the interior of the territory [109] is also confirmed, as well as in the Pre-Pyrenees where many settlements feature silos of great volumetric capacity [110–112 (p. 328)].

It is during this period that the *silo fields* preserve surpluses, products for potential trade and commerce, accumulated during times of economic expansion. Initially they were concentrated along the coastal are [21, 113, 114] from where they expanded throughout the territory as of the 5[th] century BCE [115 (p. 216), 116]. This corroborates the phenomenon of the

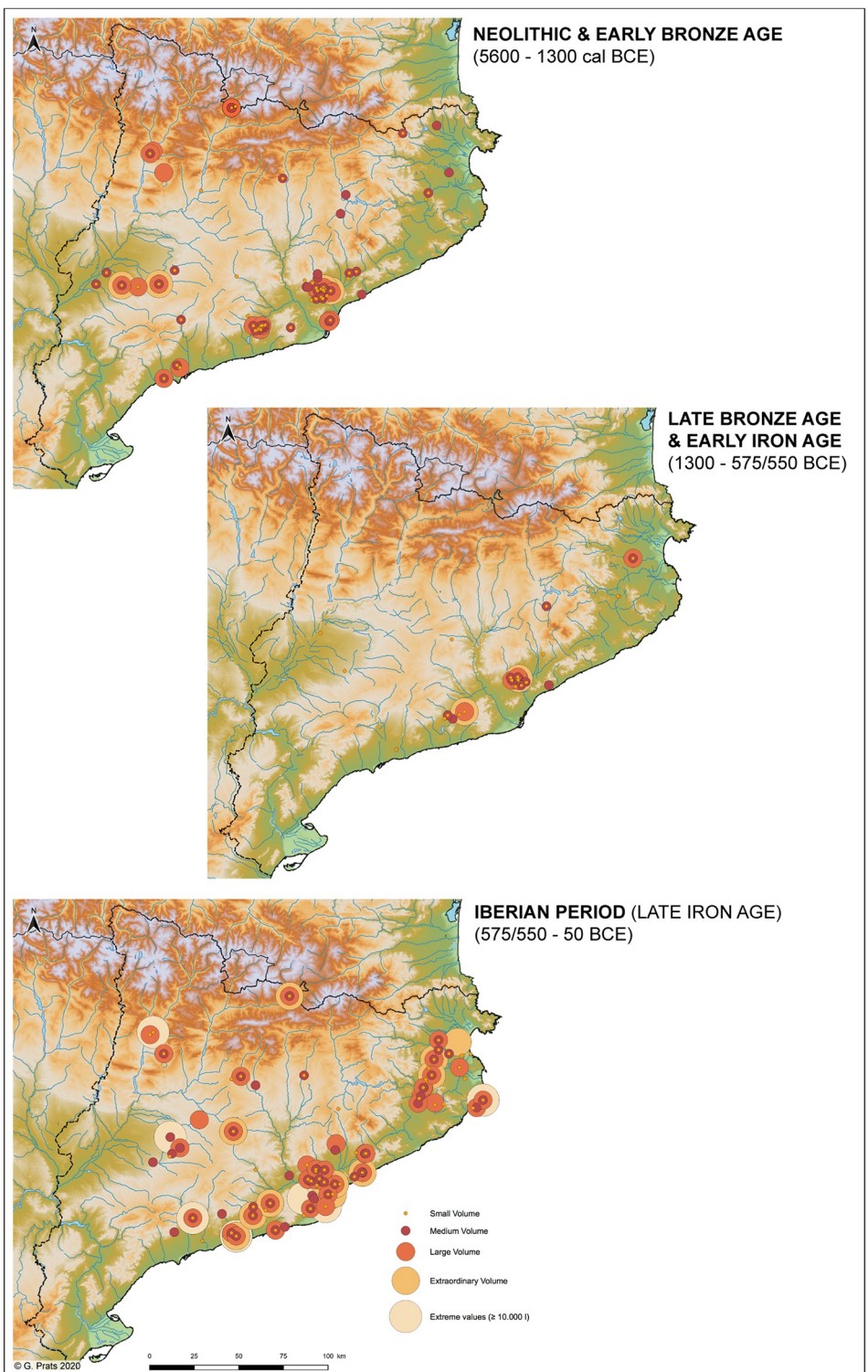

**Fig 8. Maps of the northeast of the Iberian Peninsula indicating the spread of all the sites containing small, medium large and very large silos by chronological periods (and extreme values) (© European Union, Copernicus Land Monitoring Service [2016], European Environment Agency (EEA)).**

implantation of centres in the interior of the NE of the Peninsula that accumulated and managed reserves and reveals their complex degree of organisation. Moreover, the control of resources in no case escapes the leaders of these societies, clearly evidenced by stately buildings and residential complexes [112] (p. 333). In this sense, the growth of the number of silos and their volumetric capacity in the NE of Iberia is comparable to what took place in Western and Central Languedoc stemming from contacts with the Mediterranean world [117].

Finally, a gradual abandonment of the *oppida*, centres of special prominence, and many of the *silo fields* [118] took place in the 2[nd] century BCE. Silos nonetheless did not disappear but were now distributed in new settlements. Worth highlighting, besides the intensive rural occupation of the Catalan coastline [112] (p. 323), 119], is the occupation of the plains of the inland and Pre-Pyrenees. The increase in number and concentration of silos in various areas of the territory evidence an increase in agricultural production [119]. Likewise, the increase of production of this new pattern of settlement was accompanied by the economic pressure imposed by Roman control of the territory. In this context, the new distribution of land from the mid-1st century BCE linked to the creation of cadastres and the founding of cities involved a concentration of the population and a socio-economic organisation based on new relationships of property and on an intensification of production processes [120, 121]. In any case, the archaeological record reveals a decrease in the role of silos at this time [21 (pp. 181–182), 122 (p. 132)].

## Household storage: Changes stemming from agricultural innovation

Storage capacity around and below the mean is associated with the normal agricultural production of a domestic unit. An important aspect to consider when examining the range of silo volumetric capacity from the Early Neolithic to the Early Bronze Age is the permanence of the minimum volumes (<500–1500 L). This suggests a perseverance of the habitual system of exploitation of the territory and the survival of "traditional" small and medium silos linked to each domestic unit (Fig 7). As illustrated on the distribution map (Fig 8), these features are common to almost all settlements of these periods both along the coast and in the interior of the study area. Neolithic agriculture was based on the cultivation of a wide variety—depending on each settlement or territory—of cereals (barley, mainly naked forms, naked wheat, emmer, einkorn), pulses (pea, lentil, fava bean, common vetch) and oleaginous (flax and opium poppy) complemented by harvesting a variety of wild fruit [123–126]. Moreover, as toiling of the land was carried out with digging sticks [127], the size of the workforce was a factor that had an effect on the extension of the surfaces to cultivate. Fields were thus smaller, permanent, and most likely managed intensively as is observed in this time frame in other areas of Europe [128]. In this sense, the diet was not exclusively based on harvested crops, but also on products collected in nature that possibly served to compensate when the agricultural production did not suffice. Larger storage features are identified since the Late Neolithic, and especially in the Early Bronze Age. Yet these are not widespread and in the study area are clearly concentrated in four specific settlements, two along the central coast and two in the interior (Fig 8). These changes stem from a transformation of the agricultural production model. The carpological record also reveals a change toward specialising in the cultivation of the naked wheat, more demanding in terms of soil nutrients, and hulled barley which is more resistant to harsh climatic conditions [129]. It is very likely in this context that the domestic units functioned autonomously and progressively developed toward a more extensive type, with annual or biennial harvests, and assisted by draught animals and the plough [130 (p. 24), 131 (p. 50)]. Although it is possible to speculate as to the existence of several contemporary silos for each

domestic unit, the study assumes the average silo capacity of 950 L to be a minimal estimate of productivity at the domestic level during this timeframe.

The analysis of the small and medium volumetric capacity reveals a substantial hike during the Late Bronze Age (minimum: 1,320 L, and mean: 2,063 L), a tendency maintained during the Early Iron Age (minimum: 1,234 L) (Fig 4). The fact of an increase of the minimum suggests a change in the economic strategy, at least at the settlements of the Catalan central coast where the silos of this chronological phase appear to be concentrated. Moreover, an abandonment appears to take place of the storage features heretofore considered small (Fig 7). These values suggest a domestic storage in structures with a capacity of up to 1,500/2,000 L, which is already above that of those of the Neolithic and the Early Bronze Age. This change is linked with the emergence of a silo type that dominates this phase: the bell-shaped silo [25, 63]. Hence this may be indicative of a process of systematisation of agricultural practices that also affects the "how is a silo made". It must be noted that this premise did not solely affect the Catalan coastal zone. Hence it is also possible that these silos adapted less to productive variability and are a less reliable reflection that those of earlier phases of poor harvests.

This phase coincides with a significant increase of the number of sites along the coastal zone of the NE of Catalonia [132] (p. 61) and, therefore, to a greater pressure on the land for a cultivation linked hypothetically to the development of the model of extensive agriculture. It is necessary, in any case, to develop further research on this question. Thus, exploiting more land required more work and the use of draught animals. This led to a widespread change in the pattern of cattle slaughter (with a focus on older animals). This could be explained, as indicated by Albizuri *et al.* [83] (p. 24), by the gradual specialisation of raising livestock and labour exploitation precisely with a more widespread application of the plough and draught animals. Agriculture was based, as in the previous phase, on cultivating winter and long-cycle cereals such as hulled barley and naked wheat. In addition, it is this time that sees the final expansion of millets [129], the introduction of spring and short-cycle crops that increase productivity through crop rotation. Likewise, evidence of the increase of pulses supports this hypothesis as they played an essential role in the cycle contributing nitrogen to the soil, crucial to cereal cultivation.

The progressive diversification of silos and their increase in volume (means and medium) in the Iberian period, coupled with a growing number of structures can be equated with an expansion and diversification of the number of agrarian settlements in the NE of the Iberian Peninsula [63]. Furthermore, the economic and productive systems are clearly more and more centred developing surpluses [53]. The agriculture system of the Early Iberian period inherited from the Bronze Age and consolidated in the Early Iron Age appears to experience a turning point provoked by the "democratisation" of the means of production. This is due to access to certain iron agricultural tools, the potter's wheel and the introduction of the rotary quern and the Iberian pushing mill [133, 134] that led to modification of the volume and practices of domestic production [135] (p. 137).

Certain authors consider that a system of crop rotation was put into place during the Early Iberian period which ensured the maximum exploitation of the land and, consequently, an increase in production [136]. Silo volumetric capacity in this time frame follows a pattern similar to that of the previous Early Iron Age and different from that of the subsequent Middle and Late Iberian periods. This suggests a uniformisation of the traditions of consumption and/or of production among the sites of the two periods, in spite of the scarcity of data available for the Early Iberian (n. of sites and silos) (Table 1). It is therefore possible to assume that the possible reduction of the number of silos is justified by an increase in volumetric capacity. There is no evidence of this in the Middle and Late Iberian period as the increase in silo volume is

linked to an escalation in the number of structures per site and, in general, throughout the territory [63].

During the Middle Iberian period the estimate of the minimum volume attains 2,500 L with average volumes at about 4,500 L (Fig 7) and a volumetric average of 2,769 L (Fig 4). Features with these volumes are spread throughout the territory with a certain concentration along the central coast and northeast of Catalonia (Fig 8). In addition, these features are located both in rural settlements that manage their own production as well as in larger centres that accumulated surpluses [63]. Thus, the structures serving to cover the needs of the domestic units increased their minimum volume leading to a growth of their productive capacity. The necessity of storing more products, undoubtedly subjecting the territory to more pressure, is linked to a consolidation of an extensive cereal-based agriculture, an expansion of viticulture, as well as a surge in suid and bovine consumption (and their secondary products) with their slaughter taking place before their fourth year [136] (p. 85), [137]. These aspects indeed point toward an expansion leading to an increase in the capacity of sustenance of the territory. Finally, silos of small and medium volumetric capacity in the Late Iberian period diminish with respect to the preceding period (Fig 7). This suggests the limitations of cultivating larger tracts of land, a trend identified in previous periods, and reducing the number of agricultural units to several family nuclei in the same sector.

## Surpluses: From nuclear to extended families and the surge of inequality

Small storage features with a capacity of less than 1,000 L corresponding to the average production capacity of a family unit are predominant during the Neolithic [71] (p. 141). There are, as noted above, several cases in the NE of the Iberian Peninsula that surpass this volume (2000 L in the Early Neolithic; 4000 L in the Late Neolithic). It is therefore legitimate to speculate as to the existence of surpluses during this stage? If a surplus corresponds to a quantity of products beyond the immediate needs of a domestic unit (see section 1), then the Early Neolithic storage features exceeding 1,000 L fall into this category. Considering the capacities of the other silos in this period and their volumetric mean, the few cases with volumes surpassing what is considered the strict necessity for the survival of a domestic nuclear unit could theoretically equate with surpluses. They could also correspond to a modest accumulations of reserves to face times of uncertainty or shortage, or the yield of a singular harvest. It is in fact logical to store more than needed due to the uncertainty of future access to resources, as well as the risks of a decline in production stemming from infection of the stored products by animals or insects [126] or theft. In any case, what was the volume necessary to assure a regular surplus? All appears to indicate that this capacity was quite limited. The economy of these communities was thus mixed, that is, not exclusively focused on agriculture [75, 123, 138].

The economic strategy of storage in domestic environments did not change in the Early Bronze when compared to the preceding phases as most of the small and medium structures range respectively from 1,000 to 2,500 L (Fig 7). However, communities of the Early Bronze Age reveal a noteworthy increase in average storage capacity. It is possible that this phase saw a consolidation of surplus production by the domestic units, probably coupled with a greater focus on an agricultural economy and a reduction of consumption of wild resources [139]. A volumetric intra-site diversity (Fig 6) also persists as in the previous periods. Hence this does not serve as evidence of significant changes of inequality within settlements.

A consolidation of this system took place in the Late Bronze Age evidenced by both a general increase of agricultural output and surpluses. This appears to be greater than that of the previous phase as suggested by the higher frequency of maximum values (25% of silos ranging between 2,719 and 4,729 L) (Fig 4). This represents what is considered the maximum

agricultural productivity attainable by a nuclear family using the plough [8] (p. 288). In fact, the use of oxen or even horses for draught, and data throughout Europe pointing to the plough since the 4th millennium BCE, bolster the idea of the use of these techniques [49, 83 (p. 28)]. In addition, socioeconomic models in continental Europe at the end of the 2$^{nd}$ millennium BCE appear to be affected by an abundance of production linked to favourable factors (climate change, population growth, human mobility inclined to exploit new territories) that generated surpluses facilitating the exchange of products [140 (p. 60), 141].

Although silo capacity in the Early Iron Age follows the same tendency as that in the Late Bronze Age with respect to the minimum values, its maximum values increase significantly (Fig 4). Indeed, silo capacity analysis (Fig 7) indicates that all the volumetric groups increased during the Early Iron Age when compared to the Late Bronze Age. It is nonetheless the mean and large volumes that escalate the most. This suggests an expansion of domestic production and the ability to amass surpluses owing to a gradual expansion of exploiting arable land and demographic growth, elements deriving from the Late Bronze Age. This could explain the increment in agricultural production and the existence of communities perhaps already deliberately generating surpluses [132] (p. 63). A notable increase can be observed in the standard deviation at the intra-site level (Fig 6) suggesting a consolidation of silos of great volume within the same site. This could be due to the incipient development of inequality between domestic units or the outset of the development of extended families with a greater productive capacity [69].

The Iberian period is characterised by a continuity of surplus values suggesting few upheavals in the cultivation of cereals in the study area. However, two innovations led to a major change: the use iron for agricultural tools and the introduction of arboriculture. This is especially characteristic of the region of Valencia from the 5$^{th}$ century BCE onwards marked by a balance of cereals and fruit trees. The presence of fruit trees in the northeast of the peninsula, by contrast, is represented almost exclusively by viticulture [134] (pp. 7–9). All this takes place within the framework of a complex social structure founded on extended families [142] (p. 87) and a hierarchy organised in concentrated and stable settlement nuclei benefitting from long-distance commercial networks. Silos of great capacity are thus mainly found along both the coastal and pre-littoral areas and in the interior (Fig 8).

According to Zabala and Bacaria [143], the abandonment of the large *silo fields* during the Late Iberian period stems from new commercial strategies conditioned by Romanisation that led to a drastic reduction in storage capacity. In this sense, it is noteworthy that despite the fact that silo groupings are not as numerous and that the features themselves are not necessarily large, their volumetric average corresponding to 3,080 L (Fig 4) is the greatest of all the studied periods. Thus, in spite of the abandonment and disuse of specialised centres serving to amass surpluses, and the resulting reduction in silo numbers, these communities retained a considerable productive capacity [63].

## Supra-household storage: The emergence of complex societies

The boundary in the domestic unit between a normal surplus and one exceeding the habitual is blurred, in particular in the initial stages of the Neolithic. How can one interpret storage capacity of over 2000 L in this early period? Could it represent the ability of certain individuals or households to accumulate a volume beyond their basic needs? The origin of surplus agricultural production in the Neolithic is linked to the exchange of goods of prestige [144]. Thus, the pattern of open-air settlements and the development of networks of medium and large-scale circulation suggests a complex economic model marked by examples of labour specialisation oriented toward exchange [145]. In fact, this period's funerary record and the differences in

the grave goods indicate the likely existence of social inequality [146]. Silo maximum capacity since the Late Neolithic in isolated cases increases and could evidence specific needs in the context of episodes of social grouping or indicate collective use of the structures by the community's inhabitants. Certain phenomena such as the appearance of decorated menhirs or large stelae [147] (pp. 269–284), often associated with moments of social coalescence, could bolster the first of these theories.

Noteworthy are the findings from the recent excavations of Neolithic sites to the south of the Ebro, specifically in the area of Valencia. Characteristic of their silos dating between the 5[th] and the end of the 3[rd] millennium BCE are the large models exceeding 10,000 L [62, 71]. These values, totally unheard of in the study area, suggest that certain Neolithic communities were capable of accumulating large amounts of agricultural products [49]. Judging from the available data, these surpluses undoubtedly correspond to the cumulative efforts of several domestic units, which, according to the authors, could be explained as an indicator for the appearance of local social hierarchies [147].

The Early Bronze Age is the first period to systematically betray silos with exceptional capacity. Two cases exceed 4,000 L and another two exceed 5,000 L, unusual values for this time frame. These volumes, despite the homogeneity of the data, are atypical. The systematic construction through time of larger features with a greater capacity could respond to a need for collective (not individual) storage. Although it is not possible to rule out that these features represent the outset of modest speculative and commercial actions, they could have a cooperative finality. Certain authors suggest that they could reflect to the beginning of social inequality as part of the domestic agricultural production was channelled in favour of a few individuals or the collective [95] (p. 303). This could be in the hands of a personality generally identified as a *Big man* that redistributed the products. These individuals held a prominent and prestigious status, albeit not hereditary, recognised by the rest of the community.

The presence of large and very large volumes (4,000–7,000 L) at very few sites during the Early Iron Age (Fig 8) suggests the development of surplus management in the hands of emerging elites, that is, a minority that presided over the exchange and redistribution of the surpluses. This scenario is linked to a new order of settlements [53, 148 (p. 261)] and the emergence of large concentrations of silos known as *silo fields*, features that were not only destined for domestic storage. Their great number reveal a capacity of gathering a surplus generated by the community and potentially the production of other small nuclei. They thus represent supra-household storage designed for speculative reasons (Fig 8), an economic strategy subsequently fully developed during the Middle Iberian and a precedent of the future *silo fields* of Iberian settlements [72] (pp. 40–41).

This increase in productivity can be linked to an external demand and a progressive increase of imported goods in the final chronological phases under study [53 (p. 128), 149 (p. 180)]. In addition, according to Hinojo and López [150] (p. 142), the emergence at this time of settlements with large groups of *silo fields* serving to store agricultural surpluses demonstrates the success of these crops and the subsequent cereal-specific specialisation of certain areas of the NE of the Iberian Peninsula [134].

Consolidation of the ability to generate supra-household storage takes place during the Iberian period. Very large silos surpassing 10,000 L, in fact, are mainly concentrated in nuclei in eastern Catalonia and the central pre-littoral and northeastern zones (Fig 8). In fact, many of these sites are defined as centres concentrating surpluses of grains originated from bordering or remote areas. This scenario suggests a type of social inequality where part of the agricultural production of domestic units was remitted to another and/or part of the production of certain nuclei was ceded to more important and powerful centres.

Surpluses to meet the commercial needs at that moment are reflected by volumes exceeding 7,000 and 10,000 L (Fig 7). Sites with silos of this capacity, few in number, are found both along the coast and in the northern interior of the territory (Fig 8). These features disassociate themselves from all the others. Silos with a capacity exceeding 40,000 L during the Middle Iberian period are also noteworthy. Although not depicted in the box plot (Fig 4), they do appear among the spread of extreme silos on the map of Fig 8. To these can be added other huge silos with capacities between 50,000 L and 60,000 L, and, in one extreme case up to 80,000 L [151] (p. 25). Their immense dimensions indicate they are collective features serving for commercial speculation, notably maritime trade throughout the Mediterranean [52]. Their respective settlements and elites charged with their management organised and exploited the surrounding territory. In general, their capacity during the Late Iberian period is much homogeneous than in the Middle Iberian. In spite of the fact that they still reveal extreme values (Fig 4), they are subordinate to those of the previous period. This could stem from an intensification of Romanisation throughout the territory which could have led to a decrease in the capacity of concentrating and hoarding cereals by the Iberian elite (Olesti 2007: 123, in [108] (p. 82)].

## Conclusion

The current analysis falls within an exceptional spatio-temporal framework ranging from the appearance and consolidation of agricultural practices in the NE of the Iberian Peninsula to the emergence of the first cities followed by an integration of the communities into the Roman system. The choice of this type of approach is key as it offers to diachronic perspective of the changes in storage throughout different socioeconomic and political systems.

This study distinguishes three levels of storage of agricultural products: household, surplus and supra-household. The first is equated with a more or less constant strategy throughout the Neolithic and the outset of the Bronze Age with few changes in social structure and agricultural productivity at the domestic level. The data suggest, nonetheless, an increasingly important role of cereals. A surge takes place as of the Late Bronze Age, period of technological changes and socioeconomic models linked to a more extensive agriculture aimed at maximising production. The use of draught animals and a probable increase in the number of members of the family led to greater exploitation of arable land that will continue to expand through the Iberian period.

The data offered by underground silos indicate that the capacity to generate agricultural surpluses by Neolithic communities was modest, limited by their workforce. Attaining surpluses throughout the subsequent Bronze Age, in turn, is common within the domestic unit. Surpluses increase in the Iron Age due to the gradual extension of cultivable land and use of draught animals. Hence, attaining surpluses will become a clear objective of the communities of the NE of the Iberian Peninsula. A continuity is observed throughout the Iberian Culture in terms of surpluses stemming from the consolidation of a highly productive agricultural model.

Finally, certain storage features exceed by far what is considered domestic production. Evidence of supra-household storage first appears at the end of the Neolithic and the beginning of the Bronze Age. This corresponds to a period when it was not simple to generate surpluses and reveals that the community adopted collective strategies that could reflect the emergence of social inequalities. Supra-household storage during the Middle Iberian takes on a very different form evidenced by the vast *silo fields* characterised by features of inordinate dimensions. The accumulation of agricultural surpluses and their distinct form of management by certain social groups stems from an internal development of Iberian society, motivated by *input* from Mediterranean contacts. The intent of these mega features is totally commercial and speculative. In addition, they coincide with a specialisation of cereal production and the existence of

true networks of territorial control in the framework of a complex social and political structure.

In short, the findings of this study shed light both on the question of storage practices in the NE of the Iberian Peninsula and the Western Mediterranean as well as on agricultural practices and productions. It is our hope that this study's model be applied to other regions and areas where this type of analysis has yet to take place. New research from different social and temporal levels should lead to develop new approaches toward the question of storage and its role in the sociocultural evolution of past communities.

## Supporting information

**S1 Text. Definitions.**
(DOCX)

**S1 Table. Sites list and type of settlement.**
(DOCX)

**S2 Table. List of Ø/D index ranges and the total number of pits for each range and chronological period.**
(DOCX)

**S3 Table. Silo-pit list, chronology (\* see Table 1), site name and volumetric calculation.**
(DOCX)

## Acknowledgments

This study derives from the doctoral thesis of G. Prats [63] (University of Lleida). Site reports were consulted at the Archaeological Service of the Generalitat of Catalonia. Other reports were consulted at different archaeological institutions and companies. We wish to thank all the institutions and individuals for their assistance and access to their data. We also wish to thank the many archaeologists and researchers for generously providing unpublished information, and to Héctor Matínez for the graphs in R. The translation is by T. J. Anderson.

## Author Contributions

**Conceptualization:** Georgina Prats, Ferran Antolín.

**Data curation:** Georgina Prats.

**Formal analysis:** Georgina Prats.

**Funding acquisition:** Ferran Antolín.

**Investigation:** Georgina Prats.

**Methodology:** Georgina Prats.

**Project administration:** Ferran Antolín.

**Resources:** Georgina Prats.

**Supervision:** Ferran Antolín, Natàlia Alonso.

**Validation:** Georgina Prats.

**Visualization:** Georgina Prats.

**Writing – original draft:** Georgina Prats.

**Writing – review & editing:** Georgina Prats, Ferran Antolín, Natàlia Alonso.

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
