## [Decision Letter · Decision Letter 0]

6 May 2020

PONE-D-20-01811

Household storage, surplus and supra-household storage in Prehistoric and Protohistoric societies of the western Mediterranean

PLOS ONE

Dear Mrs. Prats,

Thank you for submitting your manuscript to PLOS ONE. After careful consideration, we feel that it has merit but does not fully meet PLOS ONE’s publication criteria as it currently stands. Therefore, we invite you to submit a revised version of the manuscript that addresses the points raised during the review process.

All comments need to be addressed before re-submission.

We would appreciate receiving your revised manuscript by Jun 19 2020 11:59PM. To enhance the reproducibility of your results, we recommend that if applicable you deposit your laboratory protocols in protocols.io, where a protocol can be assigned its own identifier (DOI) such that it can be cited independently in the future. For instructions see: http://journals.plos.org/plosone/s/submission-guidelines#loc-laboratory-protocols

We look forward to receiving your revised manuscript.

Kind regards,

Peter F. Biehl, PhD

Academic Editor

PLOS ONE

Additional Editor Comments:

Your manuscript has now been seen by a referee, whose comments are appended below. You will see from these comments that while the referee finds your work of great interest, the reviewer has raised come concerns that must be addressed. Please re-submit after all the comments have been addressed.

Journal Requirements:

2. a) We note that Figure 1 in your submission contain copyrighted images. All PLOS content is published under the Creative Commons Attribution License (CC BY 4.0), which means that the manuscript, images, and Supporting Information files will be freely available online, and any third party is permitted to access, download, copy, distribute, and use these materials in any way, even commercially, with proper attribution. For more information, see our copyright guidelines: http://journals.plos.org/plosone/s/licenses-and-copyright.

  b)

We note that Figures 2 and 8 in your submission contain map images which may be copyrighted. All PLOS content is published under the Creative Commons Attribution License (CC BY 4.0), which means that the manuscript, images, and Supporting Information files will be freely available online, and any third party is permitted to access, download, copy, distribute, and use these materials in any way, even commercially, with proper attribution. For these reasons, we cannot publish previously copyrighted maps or satellite images created using proprietary data, such as Google software (Google Maps, Street View, and Earth). For more information, see our copyright guidelines: http://journals.plos.org/plosone/s/licenses-and-copyright.

You may seek permission from the original copyright holder of Figures 2 and 8 to publish the content specifically under the CC BY 4.0 license. 

If you are unable to obtain permission from the original copyright holder to publish these figures under the CC BY 4.0 license or if the copyright holder’s requirements are incompatible with the CC BY 4.0 license, please either i) remove the figure or ii) supply a replacement figure that complies with the CC BY 4.0 license. Please check copyright information on all replacement figures and update the figure caption with source information. If applicable, please specify in the figure caption text when a figure is similar but not identical to the original image and is therefore for illustrative purposes only.

"This study derives from the doctoral thesis of G. Prats (2017, University of Lleida) financed by a FI-

2011 grant from the Generalitat de Catalunya and linked to projects HAR2008-0526 and HAR2016-

78277-R. Composition of the text was funded by the Swiss National Science Foundation project entitled

Small seeds for large purposes: an integrated approach to agricultural change and climate during the

Neolithic in Western Europe (AgriChange) (PP00P1_170515, PI: Ferran Antolín). Site reports were

consulted at the Archaeological Service of the Generalitat of Catalonia.

"This research has been funded by the Swiss National Science Foundation as part of a

SNF Professorship (PI: F. Antolín), grant number: PP00P1_170515, and by the

projects HAR2008-0526 and HAR2016-78277-R."

Reviewers' comments:

Reviewer's Responses to Questions

**Comments to the Author**

1. Is the manuscript technically sound, and do the data support the conclusions?

Reviewer #1: Yes

2. Has the statistical analysis been performed appropriately and rigorously? 

Reviewer #1: Yes

3. Have the authors made all data underlying the findings in their manuscript fully available?

Reviewer #1: No

4. Is the manuscript presented in an intelligible fashion and written in standard English?

Reviewer #1: Yes

5. Review Comments to the Author

Reviewer #1: As you are well aware, the preservation at La Draga is in all ways exceptional, so we should expect exceptional findings there what no means that spikes could be stored elsewhere but not preserved.

I haven’t been able to see the ai file containing figure 2 (the tp_map2010.jpg is linked to the figure but not provided), anyhow with the low resolution one, maybe is a good idea to use categorical symbols by periods to show how the sites are located in the area, or by any other category, in this way this figure can provide complementary information.

This will be one of my little objections, I miss the spatial component in the study both at the intra-site level (what is remarked by the authors) and to the regional level. Are bigger capacity sites close to the coast or main rivers? The information is on the paper, but on my view should be more explicit, maybe just adding main rivers to the maps will help the reader.

I don’t see why “all features devoid of clear chronocultural contexts were discarded” at least for a general approach they can be useful.

I’ve found some problems with the (top) diameter/depth correlation, because I am afraid that a good number of structures lack their uppermost part due to taphonomic processes including both manual and mechanical plowing as well modern machinery removal as far as most of the sites have been location when carrying public works as far as I understood. As you implicitly recognize citing a Bogaard “The calculations are therefore an underestimation of their real storage capacity”, on my view this underestimation is due to the fact that upper part is gone, so, makes it sense to use the top diameter for any index?

This D/Ø index is nonetheless only applied to features starting with the Early Bronze Age because if applied to the Neolithic structures most would have to be discarded.

In the supporting information all the measures used to calculate indexes or volumes should be included for the sake of reproducibility.

This formula (1 cm3 = 0 1 ml) is quite elemental and have an extra 0, 1 cm3 = 1 ml, maybe is not needed.

Rstudio is an integrated development environment for R that needs of libraries as you perfectly Know, then you should mention what library is used to draw the boxplots, graphics? If so maybe the option notch=TRUE will enhance the result in the lines of the violin plots presented. These ones are more problematic because violin plots are not part of the basic R, some extra libraries are needed, which one did you choose? violplot? In any case, both graphics [1] an violplot [2], or any other choice for this matter, deserve to be cited. In the same way the options used to calculate SD boxplots should be indicated because this is doable but it’s not the default option of the boxplot function in the graphics library.

I miss the total number of silos in each phase, it will be helpful to know if the anomalies in the Early Bronze Age when compare with Late Bronze Age are reflecting a small sample or not.

You say “The Middle and Late Iberian period reveal greater differences between the extremes

(minimum and maximum) and, therefore, are the periods with the most dispersed and variable data” but it seems to me that this is true for the Early Bronze Age where the range seems similar to the Early Iron Age.

I have no doubts about your statements about the 50% (the values between the first and the third quartile of your calculations [3]) of silos capacity being between say “Late Bronze Age is between 1,320 and 2,719 L,… Early Iberian between 1,414 and 3,629 L” but in the next sentence you move from the first and third quartile to the maximum (outliers included?) of this two phases “reveals the greatest increase with values attaining of 4,729 L (Late Bronze Age) and 6,592 L (Early Iberian)” maybe so, but the median value of the Late Bronze Age is slightly higher than this of the Early Iberian. On my view you should explain that more carefully, even if I see your point I’m not sure the data support it. Along the same line, the are sentences that are confusing, at least to me, “Atypical values appear throughout the entire Neolithic and the Early Bronze Age that are beyond the maximum range”, can any value be beyond the maximum? Instead of atypical I would suggest the use of outliers or extreme values.

‘Bronze Age and consolidated in the Early Iron Age appears to experience a turning point provoked by the “democratisation” of the means of production. This is due to access to certain iron agricultural tools’. I’m not sure of how huge is the difference between having iron tools and other tools but I guess is more important to have animals to work the land.

Maybe some comparison between silos and houses could help to gain a better insight into ‘average household’. Along the whole article it seems that there is a direct relationship between households and silos, the possibility of communal silos should be further explored beyond passing from nuclear to extended families.

In the region of Valencia, besides the references of Pascual and Pérez Jordá, there are some papers about both silos [4] and settlement size [5] that could be pertinent to introduce in the discussion, mainly because the social hierarchies are viewed from a very different view. In the Valencia region, it has been proposed the existence of hierarchies from the very beginning of the Neolithic ~5700 BC [6].

The household concept is far more complex that what is presented throughout the paper [8,9,10].

L. Thissen [10] was talking about the existence of special purpose sites, with an emphasis on storage and accumulation of resources for the first Neolithic that should be considered at a regional level in each one of the phases analyzed.

[1] R Core Team (2019). R: A language and environment for statistical computing. R Foundation for Statistical Computing, Vienna, Austria. URL https://www.R-project.org/.

[2] Daniel Adler and S. Thomas Kelly (2019). vioplot: violin plot. R

package version 0.3.4 https://github.com/TomKellyGenetics/vioplot

[3] Hyndman, R. J. and Fan, Y. (1996) "Sample quantiles in statistical packages," American Statistician 50, 361–365

[4] Gómez Puche, M., & Diez Castillo, A. (2004). El Yacimiento de Colata (Valencia, España) y los ‘poblados de silos’ en la fachada mediterránea de la Península Ibérica. Do Epipaleolitico ao Calcolitico na Península Ibérica. Actas do IV congresso de arqueologia peninsular. 235-247

[5] Bernabeu Auban, J., Molina Balaguer, L., Diez Castillo, A., & Orozco Köhler, T. O. (2006). Inequalities and Power. Three Millennia of Prehistory in Mediterranean Spain (5600—2000 cal BC). In Social Inequality in Iberian Late Prehistory, BAR international series (pp. 97-116). Archaeopress Oxford.

[6] Bernabeu Auban, J., Orozco Köhler, T., Díez Castillo, A., Gómez Puche, M., & Molina Hernández, F. (2003). Mas d’Is (Penàguila, Alicante): aldeas y recintos monumentales del Neolítico Inicial en el valle del Serpis.. Trabajos de Prehistoria, 60(2), 39-59. http://dx.doi.org/10.3989/tp.2003.v60.i2.80

[7] Tringham, R. (1991). Households with faces: the challenge of gender in prehistoric architectural remains. Engendering archaeology: women and prehistory, 93-131.

[8] Tringham, R. (2012). Households through a digital lens. New Perspectives on Household Archaeology, 81-120.

[9] Joyce, Rosemary A. "The Archaeology of Household Activities." American Antiquity, vol. 66, no. 1, 2001, p. 164

[10] Thissen, L. (2002). Time trajectories for the Neolithic of Central Anatolia. The Neolithic of Central Anatolia: Internal developments and external relations during the 9th-6th millennia cal BC. Istanbul: Ege Yayınları, 13-26. See page 20

6. PLOS authors have the option to publish the peer review history of their article (what does this mean?). If published, this will include your full peer review and any attached files.

Reviewer #1: No

---

## [Author Response · Author response to Decision Letter 0]

10 Aug 2020

June 19th, 2020

Enclosed please find the revised version of our manuscript titled Household storage, surplus and supra-household storage in Prehistoric and Protohistoric societies of the western Mediterranean by Georgina Prats (Universität Basel), Ferran Antolín (Universität Basel) and Natàlia Alonso (University of Lleida). 

We are very grateful to the Editor and the reviewer for their very helpful comments and suggestions. We have read the comments carefully, and we tried to take them into consideration.

Additional Editor Comments:

Please ensure that your manuscript meets PLOS ONE's style requirements, including those for file naming. The PLOS ONE style templates can be found at https://journals.plos.org/plosone/s/file?id=wjVg/PLOSOne_formatting_sample_main_body.pdf and https://journals.plos.org/plosone/s/file?id=ba62/PLOSOne_formatting_sample_title_authors_affiliations.pdf

Checked.

2. a) We note that Figure 1 in your submission contain copyrighted images. All PLOS content is published under the Creative Commons Attribution License (CC BY 4.0), which means that the manuscript, images, and Supporting Information files will be freely available online, and any third party is permitted to access, download, copy, distribute, and use these materials in any way, even commercially, with proper attribution. For more information, see our copyright guidelines: http://journals.plos.org/plosone/s/licenses-and-copyright.

1.You may seek permission from the original copyright holder of Figure 1 to publish the content specifically under the CC BY 4.0 license.

2. If you are unable to obtain permission from the original copyright holder to publish these figures under the CC BY 4.0 license or if the copyright holder’s requirements are incompatible with the CC BY 4.0 license, please either 

i) remove the figure or 

ii) supply a replacement figure that complies with the CC BY 4.0 license. Please check copyright information on all replacement figures and update the figure caption with source information. If applicable, please specify in the figure caption text when a figure is similar but not identical to the original image and is therefore for illustrative purposes only.

Figure 1 is from FONDAZIONE FEDERICO ZERI and it complies with the CC BY 4.0 license. We have attached the proof that the owner of that content has approved of the CC BY license being applied to their content. Here you have the link of the picture and at the end the link to Creative Commons (we are free to share — copy and redistribute the material in any medium or format).

http://catalogo.fondazionezeri.unibo.it/scheda/fotografia/41799/Anonimo%20-%20Cozzarelli%20Guidoccio%20-%20sec.%20XV%20-%20Immagazzinamento%20del%20grano%20a%20Siena%20-%20insieme

2. b) We note that Figures 2 and 8 in your submission contain map images which may be copyrighted. All PLOS content is published under the Creative Commons Attribution License (CC BY 4.0), which means that the manuscript, images, and Supporting Information files will be freely available online, and any third party is permitted to access, download, copy, distribute, and use these materials in any way, even commercially, with proper attribution. For these reasons, we cannot publish previously copyrighted maps or satellite images created using proprietary data, such as Google software (Google Maps, Street View, and Earth). For more information, see our copyright guidelines: http://journals.plos.org/plosone/s/licenses-and-copyright.

– You may seek permission from the original copyright holder of Figures 2 and 8 to publish the content specifically under the CC BY 4.0 license. 

We recommend that you contact the original copyright holder with the Content Permission Form (http://journals.plos.org/plosone/s/file?id=7c09/content-permission-form.pdf) and the following text: “I request permission for the open-access journal PLOS ONE to publish XXX under the Creative Commons Attribution License (CCAL) CC BY 4.0 (http://creativecommons.org/licenses/by/4.0/). Please be aware that this license allows unrestricted use and distribution, even commercially, by third parties. Please reply and provide explicit written permission to publish XXX under a CC BY license and complete the attached form.”

– If you are unable to obtain permission from the original copyright holder to publish these figures under the CC BY 4.0 license or if the copyright holder’s requirements are incompatible with the CC BY 4.0 license, please either i) remove the figure or ii) supply a replacement figure that complies with the CC BY 4.0 license. Please check copyright information on all replacement figures and update the figure caption with source information. If applicable, please specify in the figure caption text when a figure is similar but not identical to the original image and is therefore for illustrative purposes only.

The data used to create both maps is from the project Copernicus Land Monitoring Service (https://www.copernicus.eu/en). Access to data is based on a principle of full, open and free access as established by the Copernicus data and information policy Regulation (EU) No 1159/2013 of 12 July 2013. Bibliographic reference added in the Bibliography. Here, we send you two links if you would like to check all of the information related to permissions. 

https://land.copernicus.eu/faq/about-data-access

https://land.copernicus.eu/terms-of-use

3. Thank you for stating the following in the Acknowledgments Section of your manuscript:

"This study derives from the doctoral thesis of G. Prats (2017, University of Lleida) financed by a FI-2011 grant from the Generalitat de Catalunya and linked to projects HAR2008-0526 and HAR2016-78277-R. Composition of the text was funded by the Swiss National Science Foundation project entitled Small seeds for large purposes: an integrated approach to agricultural change and climate during the Neolithic in Western Europe (AgriChange) (PP00P1_170515, PI: Ferran Antolín). Site reports were consulted at the Archaeological Service of the Generalitat of Catalonia.

"This research has been funded by the Swiss National Science Foundation as part of a SNF Professorship (PI: F. Antolín), grant number: PP00P1_170515, and by the projects HAR2008-0526 and HAR2016-78277-R."

The current Funding Statement is ok and we have removed the funding-related text from the manuscript. 

4. Please include captions for your Supporting Information files at the end of your manuscript, and update any in-text citations to match accordingly. Please see our Supporting Information guidelines for more information: http://journals.plos.org/plosone/s/supporting-information.

Captions for the Supporting Information included at the end of the manuscript.

5. Review Comments to the Author

Please use the space provided to explain your answers to the questions above. You may also include additional comments for the author, including concerns about dual publication, research ethics, or publication ethics. (Please upload your review as an attachment if it exceeds 20,000 characters).

Reviewer #1: 

As you are well aware, the preservation at La Draga is in all ways exceptional, so we should expect exceptional findings there what no means that spikes could be stored elsewhere but not preserved. 

We checked the sentence and clarified it. 

I haven’t been able to see the ai file containing figure 2 (the tp_map2010.jpg is linked to the figure but not provided), anyhow with the low resolution one, maybe is a good idea to use categorical symbols by periods to show how the sites are located in the area, or by any other category, in this way this figure can provide complementary information. 

We already provided the ai file of figure 2. This must be some issue with the review platform.

We appreciate the reviewer’s suggestion. It would be impossible to represent 300 sites in a location map. This figure aims to show the study area and the general distribution of the sites. 

Figure 8 already shows the distribution of the sites per phase. We added some geographical information to make the map more useful, as the reviewer suggested.

This will be one of my little objections, I miss the spatial component in the study both at the intra-site level (what is remarked by the authors) and to the regional level. Are bigger capacity sites close to the coast or main rivers? The information is on the paper, but on my view should be more explicit, maybe just adding main rivers to the maps will help the reader.

We added the main rivers to the maps to make them easier to read. 

I don’t see why “all features devoid of clear chronocultural contexts were discarded” at least for a general approach they can be useful.

With this sentence, we mean that we discarded features labelled as i.e. Prehistoric or Neolithic in general. These features cannot be associated to any period, and they were not useful for the study.

I’ve found some problems with the (top) diameter/depth correlation, because I am afraid that a good number of structures lack their uppermost part due to taphonomic processes including both manual and mechanical plowing as well modern machinery removal as far as most of the sites have been location when carrying public works as far as I understood. As you implicitly recognize citing a Bogaard “The calculations are therefore an underestimation of their real storage capacity”, on my view this underestimation is due to the fact that upper part is gone, so, makes it sense to use the top diameter for any index? 

This D/Ø index is nonetheless only applied to features starting with the Early Bronze Age because if applied to the Neolithic structures most would have to be discarded.

The reviewer is right when pointing out that a good number of structures has lost their upper part because of different taphonomic processes and we mention this in the text.

We use the D/Ø index of the features as a standard criterion used in many studies which are focused on underground features. The top diameter and depth are measurements that we can extract from all negative features (not only from silo pits) and are therefore useful for a first selection of deeper pits (depth is the main characteristic of a long-term storage pit). But it is not true that we only use it from the EBA onwards. This is well explained in the text under the section “Resources and Dataset” and Fig. 3 shows how we used this index for each phase. With a database of 5000 negative structures, decisions need to be taken and they must be objective. 

This indicator can differentiate deep narrow pits from wide shallow depressions which are excluded from the study because we cannot calculate and interpret their original shapes and volumes. 

In the supporting information all the measures used to calculate indexes or volumes should be included for the sake of reproducibility.

We agree with the open access policy of PLOS ONE and that is why we chose this journal for our paper but the data shared must also be reasonable. The supplementary materials already provide an immense amount of unpublished data (all pit numbers and associated volume for each site and associated chronology) that has not yet been fully exploited by the authors. We want to emphasize that pit numbers are not even used in the paper but we share the full dataset precisely for the sake of transparency. This is already a lot more than usual where, for instance, grouped samples per site are presented in totals or the individual references of analysed material (lithics, pottery, etc.) are not made public and thus no one can ever reconstruct which elements were exactly analysed. The data we added is all that is needed to reproduce the analysis performed in the paper. 

We also added a new table S2 Table as supplementary material to reproduce Figure 3. But we cannot provide the diameter and depth of 5000 features, which is what was used to select the deeper pits. The paper only deals with the deeper pits. 

This formula (1 cm3 = 0 1 ml) is quite elemental and have an extra 0, 1 cm3 = 1 ml, maybe is not needed.

We agree. It has been deleted. 

Rstudio is an integrated development environment for R that needs of libraries as you perfectly Know, then you should mention what library is used to draw the boxplots, graphics? If so maybe the option notch=TRUE will enhance the result in the lines of the violin plots presented. These ones are more problematic because violin plots are not part of the basic R, some extra libraries are needed, which one did you choose? violplot? In any case, both graphics [1] an violplot [2], or any other choice for this matter, deserve to be cited. In the same way the options used to calculate SD boxplots should be indicated because this is doable but it’s not the default option of the boxplot function in the graphics library.

Libraries used have been added. And the reference related to both boxplots and violin plot was already cited in the bibliography (RStudio Team 2018. RStudio: Integrated Development for R. RStudio, Inc., Boston, MA URL http://www.rstudio.com/.). 

> RStudio.Version()

$citation

To cite RStudio in publications use:

 RStudio Team (2018). RStudio: Integrated Development for R. RStudio, Inc., Boston, MA URL http://www.rstudio.com/.

Regarding this reviewer sentence “In the same way the options used to calculate SD boxplots should be indicated because this is doable, but it’s not the default option of the boxplot function in the graphics library.” we should say that SD calculations for the boxplot were not made using Rstudio, they were calculated separately and the data provided in the supplementary material allows for this calculation to be reproduced. 

I miss the total number of silos in each phase, it will be helpful to know if the anomalies in the Early Bronze Age when compare with Late Bronze Age are reflecting a small sample or not.

We agree with the reviewer’s comment. We added the total number of silos in both boxplots. 

You say “The Middle and Late Iberian period reveal greater differences between the extremes (minimum and maximum) and, therefore, are the periods with the most dispersed and variable data” but it seems to me that this is true for the Early Bronze Age where the range seems similar to the Early Iron Age.

We don’t agree with this comment and is probably a misunderstanding. It is obvious that the values from the Early Bronze Age are closer to the Neolithic ones and that the Early Iron Age clearly has greater differences between the extremes than the Early Bronze Age.

The Middle and Late Iberian period reveal greater differences between the extremes, and it is also highlighted in the SD boxplot Fig. 6.

I have no doubts about your statements about the 50% (the values between the first and the third quartile of your calculations [3]) of silos capacity being between say “Late Bronze Age is between 1,320 and 2,719 L,… Early Iberian between 1,414 and 3,629 L” but in the next sentence you move from the first and third quartile to the maximum (outliers included?) of this two phases “reveals the greatest increase with values attaining of 4,729 L (Late Bronze Age) and 6,592 L (Early Iberian)” maybe so, but the median value of the Late Bronze Age is slightly higher than this of the Early Iberian. On my view you should explain that more carefully, even if I see your point I’m not sure the data support it. 

We agree and deleted the sentence.

Along the same line, the are sentences that are confusing, at least to me, “Atypical values appear throughout the entire Neolithic and the Early Bronze Age that are beyond the maximum range”, can any value be beyond the maximum? Instead of atypical I would suggest the use of outliers or extreme values.

Atypical values changed by outliers when referring to boxplots and extreme values for values above 10,000 L.

‘Bronze Age and consolidated in the Early Iron Age appears to experience a turning point provoked by the “democratisation” of the means of production. This is due to access to certain iron agricultural tools’. I’m not sure of how huge is the difference between having iron tools and other tools but I guess is more important to have animals to work the land.

In this case, we mention the means of production because we are talking about the Early Iberian period and at that moment iron tools and Iberian querns are crucial for domestic production. Metal tools allow deeper and penetration and also the ploughing of heavier soils (iron ploughshares), thus it is possible to expand arable land. We decided to keep the sentence like this (The agriculture system of the Early Iberian period inherited from the Bronze Age and consolidated in the Early Iron Age appears to experience a turning point provoked by the “democratisation” of the means of production. This is due to access to certain iron agricultural tools, the potter's wheel and the introduction of the rotary quern and the Iberian pushing mill (Alonso & Frankel 2017; Alonso & Pérez-Jordà 2019) that led to modification of the volume and practices of domestic production (Alonso 2000: 137).).

It is well known that the use of animals as workforce was essential to increase productivity, as suggested the reviewer, and we mention it in the previous paragraph. (Thus, exploiting more land required more work and the use of draught animals. This led to a widespread change in the pattern of cattle slaughter (with a focus on older animals). This could be explained, as indicated by Albizuri et al. (2011: 24), by the gradual specialisation of raising livestock and labour exploitation precisely with a more widespread application of the plough and draught animals).

Maybe some comparison between silos and houses could help to gain a better insight into ‘average household’. Along the whole article it seems that there is a direct relationship between households and silos, the possibility of communal silos should be further explored beyond passing from nuclear to extended families.

This comment shows several misunderstandings.

On the one hand, it is methodologically impossible to relate silos and houses in Prehistory in the study area because well preserved settlements are scarce and, often, the only recovered structures are the negative ones. It is also clearly not the goal of this paper, with 300 sites and more than 1500 silo pits to go down to the site scale analysis. We added this remark to the text to make it clearer.

It is not true that we make a direct equivalence between silos and households. The paper precisely deals with this issue and with how to interpret silo capacities depending on the distribution of the values in each time period. The discussion and introduction are organized in such a way that it is clear what we interpret as the average productive capacity of a household and how we identify values that go further and therefore suggest other practices. Our results are in line with ethnographic observations presented in the introduction.

The article is continuously maintained at a regional level to generate robust interpretations. We do not work with single silo values. We work with average values obtained from hundreds of silo pits.

We consider the possible existence of communal silos when values much larger than the average appear. If a specific site has a communal silo that does not exceed the average capacity of the period, it will be invisible in our analyses. This is true and it is something we need to accept in big data approaches. Such a scale of analysis would also require detailed intra-site spatial analysis, which is completely out of the scope of the paper.

As we focus on major changes from the global analysis, we have to give up the possibility of being able to identify sporadic cases of collective storage.

Communal silos do not have a direct relationship with extended families. A group of nuclear households may have a communal silo as well. We only consider the existence of extended families when the average capacity of silos starts surpassing significantly the known production capacity of a nuclear household (from recent ethnographic records), namely ca. 1500-2000 L. If the average capacity of silos approaches 4000 L, such as in the Iron Age, it must mean that the labour force available is beyond that of a nuclear household.

In the region of Valencia, besides the references of Pascual and Pérez Jordá, there are some papers about both silos [4] and settlement size [5] that could be pertinent to introduce in the discussion, mainly because the social hierarchies are viewed from a very different view. In the Valencia region, it has been proposed the existence of hierarchies from the very beginning of the Neolithic ~5700 BC [6].

We appreciate the references provided by the reviewer. We incorporated some of the references mainly in the parts in which we talk about social hierarchies. 

The household concept is far more complex that what is presented throughout the paper [8,9,10].

We defined our concept of household used in our analyses in a previous publication (Prats et al. 2020) and in the supplementary materials. It is not the aim of this paper to better understand the nature of households in the past but to see if underground storage capacity can yield significant information regarding the production capacity of households (in whichever form they may have been structured in the past) in a relevant scale for a large study region. We appreciate the references given by the reviewer (they are interesting as scientific publications indeed) but they were unfortunately of little significance to improve our definitions.

L. Thissen [10] was talking about the existence of special purpose sites, with an emphasis on storage and accumulation of resources for the first Neolithic that should be considered at a regional level in each one of the phases analyzed.

We are thankful for this reference but the Neolithic in Anatolia and the Neolithic in the Iberian Peninsula have little to nothing in common regarding settlement organization and social complexity. We have considered the existence of special purpose sites when our data has allowed for it, namely from the Early Iron age onwards, when the first sites with concentrations of silo pits appear.

---

## [Editor Report · Decision Letter 1]

13 Aug 2020

Household storage, surplus and supra-household storage in Prehistoric and Protohistoric societies of the western Mediterranean

PONE-D-20-01811R1

Dear Dr. Prats,

We’re pleased to inform you that your manuscript has been judged scientifically suitable for publication and will be formally accepted for publication once it meets all outstanding technical requirements.

Kind regards,

Peter F. Biehl, PhD

Academic Editor

PLOS ONE
---

## [Editor Report · Acceptance letter]

25 Aug 2020

PONE-D-20-01811R1 

Household storage, surplus and supra-household storage in Prehistoric and Protohistoric societies of the western Mediterranean 

Dear Dr. Prats:

I'm pleased to inform you that your manuscript has been deemed suitable for publication in PLOS ONE. Congratulations! Your manuscript is now with our production department. 

Kind regards, 

on behalf of

Dr. Peter F. Biehl 

Academic Editor

PLOS ONE